# Gingers and Their Purified Components as Cancer Chemopreventative Agents

**DOI:** 10.3390/molecules24162859

**Published:** 2019-08-07

**Authors:** John F. Lechner, Gary D. Stoner

**Affiliations:** 1Retired from Department of Medicine, Division of Medical Oncology, Ohio State University, Columbus 43210, OH, USA; 2Department of Microbiology and Immunology, Montana State University, Bozeman, MT 59717, USA

**Keywords:** gingerols, shogaols, paradols, zingerone, zerumbone, cancer chemoprevention, apoptosis

## Abstract

Chemoprevention by ingested substituents is the process through which nutraceuticals and/or their bioactive components antagonize carcinogenesis. Carcinogenesis is the course of action whereby a normal cell is transformed into a neoplastic cell. This latter action involves several steps, starting with initiation and followed by promotion and progression. Driving these stages is continued oxidative stress and inflammation, which in turn, causes a myriad of aberrant gene expressions and mutations within the transforming cell population and abnormal gene expressions by the cells within the surrounding lesion. Chemoprevention of cancer with bioreactive foods or their extracted/purified components occurs primarily via normalizing these inappropriate gene activities. Various foods/agents have been shown to affect different gene expressions. In this review, we discuss how the chemoprevention activities of gingers antagonize cancer development.

## 1. Introduction

Carcinogenesis is the series of events whereby a normal cell is transformed into a neoplastic cell. Underlying this transition is continued oxidative stress, which is the imbalance between oxidants, e.g., reactive oxygen species (ROS), and their elimination by the cell’s protective phase II detoxication mechanisms. Whereas ROS in proper amounts are regulators of homeostatic signal transduction [1,2,3], during tumorigenesis, the continued oxidative stress causes aberrant expression of more than 500 genes, both in the target cells and the cells in the lesion’s microenvironment [4,5]. Exogenous oxidative stress stems from chemical and physical stimulants; the endogenous increase in ROS derives primarily from dysregulation of the mitochondria, NADPH oxidases, and peroxisome oxidation of proteins [5,6,7,8]. This continued excess ROS results in inflammation [5], which arises as a series of pathologic events, including the invasion of neutrophils, lymphocytes, and macrophages, which are sources of pro-inflammatory cytokines and extracellular ROS [9]. This oxidative stress situation sets up a chronic active state that results in cells exhibiting the hallmarks of tumor cells, i.e., genomic instability and mutation, altered epigenetic events, inappropriate gene expression, changed microRNA translation, enhanced proliferation of the initiated cells, enabling resistance to cell death, antagonism of immune surveillance, invasion and neo-vascularization [10]. Interestingly, another hallmark of a cancer cell is that the intra-cellular concentration of ROS in tumor cells is generally higher than it is in the corresponding normal cells, and further elevation of ROS levels in tumor cells is lethal [11].

Nutraceutical cancer chemoprevention is the process whereby natural (food) substances interfere with carcinogenesis. This review discusses the chemopreventative activities of ginger. There are two species of ginger that have been used for therapeutic purposes for centuries. One is *Zingiber officinale* Roscoe (true ginger), the other is *Zingiber zerumbet* Smith (bitter/shampoo ginger) [12]. The rhizomes of both species have chemopreventative activities. We will review true ginger and its compounds initially, then bitter/shampoo ginger.

## 2. True Ginger

### 2.1. Anti-Inflammatory Studies

Ginger rhizome extract has been demonstrated to reduce swelling (edema) of carrageenan-injected rat paw [13]. Choi et al. [14] tested rhizome extract for anti-inflammatory properties in lipopolysaccharide (LPS)-treated mice and found it reduced the pathological appearances of inflammation in the liver, as well as reducing the level of the circulating proinflammatory cytokines INF-γ and IL-6. NFκB activation was also inhibited, along with the expression of iNOS and COX-2.

### 2.2. Animal and In Vitro Studies with Ginger Extract

With respect to the colon, ginger extract was an effective chemopreventative if given both during and post-carcinogen treatment [15]. NFκB and TNF-α were also down-regulated in the liver of the treated animals. When the extract was provided to the animals daily by oral administration in a prostate xenograft study, the cells formed tumors more slowly [15]. The resultant smaller tumors showed reduced levels of cyclins B1, D1, and E, with increased p21 and cleaved caspase-3. The extract-treated tumors also showed extensive apoptosis. Ginger also inhibited azoxymethane (AOM)-induced intestinal carcinogenesis in rats [16].

Using cultured human lung cancer cells, Tuntiwechapikul et al. [17] found that hTERT and its up-stream regulator c-Myc were down-regulated in a dose- and time-dependent manner by ginger extract. Elkady et al. [18] reported the same results regarding hTERT and c-Myc using human breast cancer cells. They also reported that their extract killed the cells in a dose–response manner. Interestingly, non-tumorigenic breast cells were not affected. Treatment of tumor cells with ginger extract resulted in apoptosis and decreased expression of the prosurvival genes, NFκB, Bcl-X, Mcl-1 and Survivin, as well as the cell cycle regulating proteins, cyclin D1 and CDK4. In contrast, p21 expression was increased.

The effects of ginger on ovarian cells have also been evaluated. Rhode et al. [19], found that normal ovarian cells are resistant to ginger extract killing, whereas three tumor cell lines showed significant death in a time- and dose-dependent manner. Ginger treatment of the tumor cells resulted in inhibition of NFκB and diminished secretion of vascular endothelial growth factor (VEGF) and IL-8. Pashaei-Asl et al. [20] reported that ovarian tumor cells were growth inhibited and the p53 protein was up-regulated, while BCL-2 was down-regulated following treatment with ginger. Liu et al. [21] also found that p53 was a key player in ginger-extract-induced apoptosis of endometrial cancer cells through rapid phosphorylation of the Ser-15 in the protein molecule. Apoptosis was not observed in p53^neg^ cells treated with extract.

Ginger extract inhibited the production of ROS, DNA strand breaks, and cytotoxicity caused by the incubation of HepG2 hepatocarcinoma cells with aflatoxin. Additionally, ginger extract up-regulated the Nrf2/HO-1 pathway [22]. Regarding pancreatic cancer cells, Akimoto et al. [23] reported that ginger extract suppressed cell cycle progression and consequently, induced death. The extract markedly up-regulated 5′ AMP-activated protein kinase (AMPK), a positive regulator of autophagy, and inhibited mTOR, a negative autophagic regulator. The extract also suppressed tumor growth in an orthotopic model of pancreatic cancer without adverse effects on the host animal.

### 2.3. True Ginger Extract and Helicobacter

Ginger extract inhibits growth of *Helicobacter pylori* (HP) strains, including several tumorigenic CagA+ strains, in vitro [24]. Thus, the extract may contribute to chemoprevention via inhibiting the inflammation caused by HP in the gastric mucosa. In support of this notion, Gaus et al. [25] reported that ginger extract reduced the load of HP in infected Mongolian gerbils, while significantly reducing both acute and chronic mucosal and submucosal inflammation.

### 2.4. Human Chemopreventive Efficacy Studies with True Ginger Extract

Zick et al. [26] studied people at normal risk for colon cancer by evaluating eicosanoid levels in colon biopsies of individuals who ingested two grams per day of ginger extract for 30 days. The extract was well tolerated, but there was no significant change in the concentrations of the assessed pro-inflammatory molecules. Regarding their study of patients at increased risk of colon cancer [27], 20 people were randomized to ingest 2.0 g of ginger or placebo daily for 28 days. Expression per crypt distribution of Bax, Bcl-2, p21, hTERT, and MIB-1 in colorectal biopsies were measured. Relative to placebo, Bax expression in the ginger group decreased 15.6% in the whole crypts, 6.6% in the upper differentiation zone, and 21.7% in the lower proliferative zone. p21 and Bcl-2 expressions remained relatively unchanged; however, hTERT expression in the whole crypts decreased by 41.2%. MIB-1 expression was also decreased in the whole crypts.

### 2.5. True Ginger Rhizome Bioactive Constituents

The rhizome contains >400 chemical compounds [28]. The major bioreactive constituents are gingerols, shogaols, paradols, and zingerone (Figure 1). Fresh ginger contains predominately (6)-gingerol, which, in turn, is converted to (6)-shogaol upon heating or drying of the rhizome [29]. Hydrogenation transforms shogaol into paradol [30]. The other bioactive constituent is the monocyclic sesquiterpene, zingiberene [29] (Figure 1). These constituents will be reviewed in turn.

#### 2.5.1. Gingerol

##### Metabolism of Gingerol

Jiang et al. [31] reported that ingested (6)-gingerol was rapidly absorbed into rat plasma and is readily distributed to tissues. Nakazawa and Ohsawa [32] noted that approximately one-half of orally administrated (6)-gingerol was recovered in bile as glucuronide conjugates. Humans also convert the free compounds to glucuronides [33,34]. Mukkavilli et al. [35] also found glucuronides of gingerols in plasma. These authors noted [36] that the efficacy of gingerols in tumor cells was best explained by the reconversion of conjugated molecules to the free forms by β-glucuronidase, which is selectively over-expressed in the tumor tissue. This two-step process suggests an instantaneous conversion of ingested free compounds into conjugated forms, followed by their subsequent absorption into systemic circulation and reconversion to the primary molecule in the tumor cells. Their proposed model uncovers a tumor-specific mechanistic underpinning of ginger′s anticancer activity despite the fact that there are sub-therapeutic levels of the free compound in plasma.

##### Molecular Activities of Gingerols Gleaned from Animal Chemoprevention Studies

Topical application of (6)-gingerol prior to painting with 12-*O*-Tetradecanoylphorbol-13-acetate (TPA)-attenuated mouse skin papillomagenesis initiated by 7,12-Dimethylbenz(a)anthracene (DMBA). (6)-Gingerol also inhibited tumor-promoter-stimulated inflammation, TNF-α production, and activation of epidermal ornithine decarboxylase (ODC) [37,38,39]. (6)-Gingerol caused apoptosis in B[a]P-induced mouse skin tumors and this was associated with up-regulation of p53, Bax, and cytochrome-c proteins [40]. Treatment with (6)-gingerol reduced UVB-induced intracellular ROS levels, activation of caspase-3, -8, -9, and Fas expression, along with COX-2 in mouse skin. Translocation of NFκB from cytosol to nucleus was also inhibited [41]. (6)-Gingerol inhibited the enlargement of mouse ventral prostate induced by testosterone, as well as induced apoptosis characterized by nicked DNA. (6)-Gingerol also upregulated testosterone-repressed p53 expression [42].

##### Molecular Activities of Gingerols Gleaned from Cell Culture Models

The vast majority of the molecular mechanisms of gingerols have been elucidated using cell culture models. This section of our review is broken down by cell type for ease of explanation. Whereas there are mechanism commonalities, cell-type individual systems have been described. What is unknown is whether these systems are universal and simply have not been investigated in all cell types.

##### Blood-Borne Cells

(6)-Gingerol exhibited dose-dependent inhibition of NO production and significant reduction of iNOS in LPS-activated macrophages [43]. (6)-Gingerol also decreased TNF-α expression through suppression of IκBα phosphorylation, NFκB and PKC-α nuclear translocation [44]. (6)-Gingerol inhibited proliferation of primary myeloid leukemia cells and myeloid leukemia cell lines in a concentration- and time-dependent manner, while sparing normal peripheral blood mononuclear cells. (6)-Gingerol induced ROS generation in the leukemia cells by inhibiting the mitochondrial respiratory complex I, which in turn, increased the expression of the oxidative stress-response-associated miRNA, miR-27. Elevated miR-27b expression inhibited inflammatory cytokine gene expressions associated with NFκB pathway activation. The results of the xenograft experiments showed that (6)-gingerol inhibited leukemia cell proliferation and induced apoptosis in the tumors [45].

1-Dehydro-10-gingerdione (1D10G) purified from ginger suppressed LPS-induced gene expression of NFκB, TNF-α and IL-1β in macrophages [46]. 1D10G directly inhibited the catalytic activity of cell-free IKKβ. Moreover, 1D10G irreversibly inhibited cytoplasmic IKKβ-catalyzed IκBα phosphorylation. These effects of 1D10G were abolished by substitution of the Cys(179) with Ala in the activation loop of IKKβ, indicating a direct docking site of 1D10G. Therefore, 1D10G (and perhaps gingerols, in general) has therapeutic potential in NFκB-associated carcinogenesis by directly binding to the IKKβ protein [47].

##### Digestive Cells

(6)-Gingerol exhibited anti-proliferation activity, caused G1 cell cycle arrest, induced apoptosis, and cyclin D1 expression in colon cancer cells [48]. Other investigators [49,50,51] reported that (6)-gingerol treatment caused accumulation of colon cancer cells in the G2/M phase with a mild accumulation in the sub-G1 phase. Examination showed that cyclin A, cyclin B1, and CDK1 levels were diminished, while the negative cell cycle regulators, p27 and p21, were increased. In addition, (6)-gingerol treatment elevated intracellular ROS and the phosphorylation level of p53. (6)-Gingerol inhibited cell proliferation and induced apoptosis with associated activation of caspases 8, 9, 3 and 7 and cleavage of poly ADP ribose polymerase (PARP). In addition, the compound down-regulated TPA-induced phosphorylation of ERK1/2 and JNK MAP kinases and activation of AP-1. (6)-Gingerol was found to be an inhibitor of the proinflammatory enzyme leukotriene A4 hydrolase (LTA_4_H), which is highly expressed in colorectal carcinoma cells [52,53]. (6)-gingerol was examined by docking inhibitory studies and found to be a potent inhibitor of LTA_4_H activity. Other gingerols were also tested and (10)-gingerol exhibited the highest binding effect.

(6)-Gingerol was assessed [54] for its apoptotic effects in human gastric adenocarcinoma cells. It causes an increase in ROS generation, which led to a decrease in mitochondrial membrane potential and subsequent induction of apoptosis, along with deregulation of the Bax/Bcl-2 ratio at protein level and the release of cytochrome-c. Studies [55] examining the early signaling effects of (6)-gingerol and (10)-gingerol on renal cells found that there was a slow and sustained rise of [Ca^2+^]i via, as yet, unidentified mechanisms. A study [56] of the effects of (6)-gingerol on tight junction (TJ) molecules was conducted using human-pancreatic-duct-derived cancer cells. Transepithelial electrical resistance (TER) and proteins related to TJs, were measured. TER was significantly increased and claudin 4 and MMP-9 decreased. The TJ protein levels of ZO-1, occludin, and E-CAD were increased while, at the same time, NFκB/Snail nuclear translocation was suppressed.

Cultured hepatic carcinoma cells were stimulated with IL-1β to establish an in vitro inflammatory model [57]. (6)-Gingerol attenuated IL-1β-induced inflammation and oxidative stress, as evidenced by decreasing mRNA levels of IL-6, IL-8, and SAA1, along with suppression of ROS generation in these cells. In addition, (6)-gingerol reduced IL1β-induced COX2 upregulation, as well as NFκB activity. (6)-Gingerol inhibited both the proliferation and invasion of hepatoma cells in a dose-dependent manner [58]. The cells accumulated in the S phase, and exhibited an elongated doubling time and apoptosis. Hepatoma cells previously cultured with H_2_O_2_ showed increased invasive activity, and (6)-gingerol suppressed the ROS-potentiated invasive capacity. Furthermore, (6)-gingerol reduced intracellular peroxide levels in the cells, suggesting anti-oxidant activity.

##### Breast Cells

The effect of (6)-gingerol on adhesion, invasion, and motility activity was assessed by measuring the levels of MMP-2 and -9 in cultured human breast cancer cells [59]. Treatment led to a concentration-dependent decrease in these proteins. Another study [60] found that (10)-gingerol was more potent than (6)-gingerol in inhibiting growth of human and mouse mammary carcinoma cells. Further investigation of (10)-gingerol showed that it also suppressed the growth of estrogen receptor-bearing and HER2-overexpressing breast cancer cells. The inhibitory effect was associated with S phase cell cycle arrest and the induction of apoptosis associated with mitochondrial membrane permeabilization and the release of cytochrome c.

Fuzer et al. [61] investigated the effects of (10)-gingerol on breast tumor cells and non-malignant human breast cells using 3D cultures in order to mimic the in vivo microenvironment. They found selective cytotoxicity against the malignant breast cancer cells. The compound reverted the malignant phenotype, down-regulated the expression of EGFR and β1-integrin, and induced apoptosis in the cancer cells. Normal cells were not affected. Another study by this group [62] using (10)-gingerol showed a concentration-dependent induction of apoptotic death of mono-layered breast cancer cells. (10)-Gingerol, which was well tolerated by the syngeneic mice, inhibited orthotopic mammary tumor cell growth and induced a marked increase in caspase-3 activation in the tumors. Using both spontaneous and experimental cell metastasis assays [63], (10)-gingerol was found to inhibit metastasis of breast cells to multiple organs including lung, bone and brain [64]. Interestingly, inhibition of brain metastasis was observed even when treatment was initiated after surgical removal of the primary tumor. The effects of (10)-gingerol on breast cells was also investigated by Joo et al. [65]. They reported that treatment with the compound inhibited cell proliferation through down-regulation of cyclin-dependent kinases and cyclins, and induced G1 phase cell cycle arrest. In addition, (10)-gingerol treatment blocked in vitro cell invasion in response to mitogenic stimulation.

##### Skin Cells

Application of (6)-gingerol to mouse skin inhibited TPA-induced COX-2 expression through suppression of NFκB nuclear translocation. In addition, (6)-gingerol inhibited the phosphorylation of p38 MAPK [66]. Kim et al. [67] substantiated and extended these findings. Pretreatment of mouse skin with (6)-gingerol resulted in a decrease in both TPA-induced NFκB DNA binding and transcriptional activities through suppression of IκBα degradation. Phosphorylation of IκBα was substantially blocked. Moreover, (6)-gingerol prevented TPA-induced phosphorylation and catalytic activity of p38 MAPK.

##### Other Cell Types

(6)-Gingerol reduced metastasis of melanoma cells in the lungs of mice [67]. Incubation of HeLa cells with (6)-gingerol caused morphological changes, DNA degradation, depolarization of mitochondrial membrane potential, and apoptosis. The expressions of caspase 3 and PARP were increased. The cells arrested in the G0/G1 phase. Expression of the NFκB, AKT and Bcl2 genes was down-regulated whereas, the expression levels of TNF-α, Bax and cytochrome c were enhanced, and death receptors (DRs) were up-regulated. (6)-Gingerol also inhibited PI3K/AKT expression and activated AMPK to induce mTOR-mediated cell apoptosis in HeLa cells [68]. Circular dichroism studies for drug–DNA interactions with both calf thymus and nuclear DNA revealed (6)-gingerol binding with conformational changes of the DNA structure, suggesting that (6)-gingerol has potential to bind with DNA in the cell [69,70]. Human papillomavirus (HPV)-expressing E6 and E7 oncoproteins inactivates p53 through proteasomal degradation in cervical cancers. (6)-Gingerol inhibited proliferation of HPV-positive HeLa cells and reactivated p53, while increasing the level of p21, inducing DNA damage, and causing G2/M cell cycle arrest. (6)-Gingerol reduced tumor volume, tumor weight, and p53 accumulation of HeLa cell xenografts [71].

(6)-Gingerol inhibited both VEGF- and bFGF-induced proliferation of human endothelial cells and caused cell cycle arrest in the G1 phase. It also blocked capillary-like tube formation in response to VEGF, and strongly inhibited sprouting of endothelial cells in rat aorta and the formation of new blood vessels in the mouse cornea in response to VEGF [72]. Another report [73] showed inhibition of endothelial cell tube formation. (6)-Gingerol induced TRAIL-mediated apoptosis of glioblastoma cells in which it increased DR 5 levels in a p53-dependent manner, while decreasing the expression levels of survivin, c-FLIP, Bcl-2, and XIAP, and increasing Bax.

#### 2.5.2. Cancer Chemopreventive Activities of Shogaols

Gingerols decay to shogaols upon heating or drying of the *Zingiber officinale* rhizome. The vast majority of the known molecular mechanisms of shogaols have been elucidated using cell culture models. Whereas there are mechanism commonalities, cell type individual systems have been described. What is unknown is whether these are universal mechanisms that simply have not been investigated in all cell types.

##### Pharmacokinetics and Metabolism of Shogaols

Rats excreted more (6)-shogaol into feces than into urine. About 90% of the dose was absorbed from the digestive tract and most of the fecal excretion was via biliary excretion. This suggests that (6)-shogaol is mostly metabolized in the body and excreted as metabolites [74]. Mukkavilli et al. [35] found glucuronides of shogaols in plasma; the same was reported by Li et al. [31]. Chen et al. [75] reported that the mercapturic acid pathway is a major metabolic route of (6)-shogaol in mice and the thiol conjugates exist in the glucuronidated and sulfated forms in urine. Fu et al. [76] reported that (6)-shogaol’s major metabolic constituents were toxic to cancer cells in mice, while being less toxic to normal colon fibroblast cells. Zhu et al. [77] also found cell-type specificity of the cysteine-conjugated forms with low toxicity against normal colon cells, but cytotoxic for the colon cancer cells. It was further demonstrated that the cysteine-conjugated shogaols caused cancer cell death via activation of the mitochondrial apoptotic pathway through increased oxidative stress, which in turn, activated p53, down-regulated Bcl-2, and increased cytochrome c release. (6)-Shogaol is also quickly metabolized in human lung cancer cells to a cysteine-conjugated form that exhibits toxicity to cancer cells similar to the parent compound, however again, the metabolite is less toxic toward normal cells [78]. Investigations [79] showed that the toxicity to cancer cells is initiated by down modulation of intracellular content of glutathione (GSH). The subsequently generated oxidative stress activates a p53 pathway that ultimately leads to cytochrome c release, and cleaved caspases 3 and 9. A dose of 30 mg/kg of (6)-shogaol or its metabolite decreased xenograft tumor burden without any associated toxicity to the nude mouse [80].

##### Animal Initiation/Promotion Chemoprevention Studies

As for (6)-gingerol, pretreatment of mouse skin with (6)-shogaol resulted in the reduction of TPA-induced nuclear translocation of NFκB [81]. (6)-Shogaol treatment of TPA-painted skin also depressed activation of MAPK1/2, p38MPK, JNK1/2, and PI3K/AKT, which are upstream NFκB and AP-1. Furthermore, 6-shogaol significantly inhibited DMBA/TPA-induced mouse skin tumor formation, as measured by the reduced multiplicity of papillomas. The effect of (6)-shogaol on UVB-induced oxidative stress and signaling was evaluated using human epidermal keratinocytes. The compound elevated intracellular ROS levels and caused apoptotic death of the cells.

##### In Vitro Studies

As was the case with gingerols, the knowledge of molecular cancer-associated mechanisms of shogaol have been gleaned using cell cultures and some xenotransplant studies. 

##### Blood-Borne Cells

Pan et al. [82] investigated the inhibitory effects of (6)-shogaol on the induction of iNOS and COX-2 in LPS-activated murine macrophage cells. Both protein and mRNA expression of the genes were down-regulated. LPS-induced transcriptional activity of NFκB and activation of PI3K/Akt and ERK1/2 was inhibited. Shieh et al. [83] examined the growth inhibitory effects of (8)-shogaol on human leukemia cells and found it induced apoptosis in a time- and concentration-dependent manner. (8)-Shogaol caused stimulation of ROS production and depletion of glutathione, a rapid loss of mitochondrial transmembrane potential, and release of cytochrome c.

Ahn et al. [84] demonstrated that (6)-shogaol inhibits LPS-induced dimerization of TLR4, a receptor that plays a critical role in inducing innate immune and inflammatory responses including the activation of downstream signaling pathways, NFκB and COX-2. Another study [85] indicated that (6)-shogaol selectively induced apoptosis in primary leukemia cells and leukemia cell lines, but not in normal cells. Docking results suggested that (6)-shogaol binds at Ser51 of eIF2α, a key regulator in the apoptosis signaling pathway. Further, (6)-shogaol induced dephosphorylation of eIF2α and caspase activation-dependent cleavage of eIF2α. Additionally, (6)-shogaol inhibited tumor growth and induced apoptosis in a xenograft mouse model of leukemia.

##### Digestive Cells

Qi et al. [86] observed that (6)-shogaol inhibited colon cell tumor growth in a xenograft mouse model by arresting growth at the G2/M phase of the cell cycle. This was associated with up-regulation of p53, p21, and GADD45α, and down-regulation of cdc2 and cdc25A. Using p53^(−/−)^ and p53^(+/+)^ cells, they confirmed that the p53/p21 pathway is the main driver of cell cycle arrest by (6)-shogaol in colon cancer cells.

(6)-Shogaol inhibited the growth of cultured human colon cancer cells and induced apoptosis through modulation of mitochondrial functions regulated by increased levels of ROS [87]. Up-regulation of Bax, Fas, and FasL, as well as down-regulation of Bcl-2 and Bcl-XL was noted, and GADD153 mRNA and protein were induced in a time- and concentration-dependent manner. (6)-Shogaol had a strong inhibitory effect on arachidonic acid release and NO synthesis [88]. Gan et al. [89] reported that (6)-shogaol inhibited anchorage-independent colony formation. The compound caused G2/M cell cycle arrest and apoptosis characterized by caspase 3 and PARP cleavage, which was associated with down-regulation of cell cycle checkpoint proteins cdk1, cyclin B and cdc25C. Spindle assembly checkpoint proteins mad2, cdc20 and survivin were also down-regulated; thus, (6)-Shogaol inhibited dose- and time-dependent accumulation of insoluble tubulin with disrupted microtubule turnover. Ishiguro et al. [90] also showed that (6)-shogaol damages microtubules. The molecular mechanism was shown to be reaction with the sulfhydryl groups of cysteine residues in tubulin, thereby disrupting mitosis. Nrf2 was also identified as a molecular target of (6)-shogaol. Following treatment of the colon cancer cells, the intracellular GSH/GSSG ratio was initially diminished but was then elevated above the basal level. Intracellular ROS correlated inversely with the GSH/GSSG ratio. (6)-shogaol increased the expression of the Nrf2-regulated phase II metabolism target genes; AKR1B10, FTL, GGTLA4, and HMOX1. The compound also modified multiple cysteine residues of Keap1 protein, thereby augmenting Nrf2 nuclear translocation.

Poorly differentiated and p53 mutant human hepatoma cells are highly refractory to chemotherapeutic agents. However, Chen et al. [91] reported that (6)-shogaol induced apoptotic cell death in human hepatoma cells via an oxidative stress-mediated caspase-dependent mechanism. ROS over-production was followed by a severe depletion of intracellular GSH; this resulted in a significant drop in mitochondrial transmembrane potential and activation of caspases 3/7, and DNA fragmentation. Hu et al. [92] reported that (6)-shogaol induces apoptosis in human hepatocellular carcinoma cells by caspase activation and endoplasmic reticulum stress signaling. (6)-Shogaol inhibited the phosphorylation of eIF2α. Furthermore, the (6)-shogaol-mediated inhibition of tumor cell growth in mouse xenografts studies was associated with inactivation of eIF2α.

The signaling pathway by which (6)-shogaol protects hepatocarcinoma cells against H_2_O_2_-induced oxidative stress was investigated by Kim et al. [93]. (6)-Shogaol enhanced the cellular GSH level, and ARE promoter activities, as well as Nrf2 accumulation in the nucleus, JNK activation, and GCS and HO-1 protein expressions. TRAIL, which induces apoptosis in tumor cells with minimal or no effects on normal cells was up-regulated by (6)-shogaol. Nazim and Park [94] reported that (6)-shogaol exerts anti-inflammatory and anticancer properties, attenuated tumor cell propagation and induced TRAIL-mediated cell death in liver cancer cells. (6)-shogaol triggered ROS production, upregulated p53 expression and changed the mitochondrial transmembrane potential.

Weng et al. [95] evaluated the anti-invasion activity of (6)-shogaol on human hepatoma cells. The migratory and invasive abilities of TPA-treated cells were reduced in a dose-dependent manner. MMP-9 activity was decreased, whereas the expression of TIMP-1 was increased. In their second study [96], (6)-shogaol was found to regulate MMP-2/-9 transcription. Moreover, it mediated a decrease in uPA that was accompanied by up-regulation of PAI-1. (6)-Shogaol concentrations of ≥2.5 μM also inhibited the phosphorylation of MAPK and PI3K/AKT, along with nuclear translocation of NFκB and STAT3.

##### Breast Cells

Ling et al. [97] reported that (6)-shogaol inhibited TPA-stimulated breast cancer cell invasion with a dose-dependent reduction of MMP-9 gene activation. NFκB nuclear translation was also decreased through inhibition of IκB phosphorylation and degradation. In addition, the compound inhibited ERK and NFκB signaling. (6)-Shogaol inhibited constitutive phosphorylation of STAT3 and its nuclear translocation in both breast and prostate cells. (6)-Shogaol induced apoptosis as characterized by cleavage of PARP, accumulation of cells in subG1 phase, and activation of caspase-8, -9, -3. Also, (6)-shogaol caused the activation of JNK, p38 MAPK, and ERK, and suppressed the growth of the breast tumors growing as xenografts in mice [98].

(4)-Shogaol, also extracted from dried ginger, was found to inhibit breast carcinoma cell migration and invasion [99]. Furthermore, it inhibited the phosphorylation of IκB and the nuclear translocation of NF-κB/Snail. RKIP, an inhibitory molecule of IKK, was up-regulated after treatment and prolonged the inhibitory effects. Finally, (4)-shogaol effectively inhibited metastasis of s.c, injected breast adenocarcinoma cells in mice.

##### Other Cell Types

A study [100] investigating the chemoprevention effect of (6)-shogaol on human NSCLC cells showed that it inhibited cell proliferation by inducing apoptosis. It also inhibited cell survival through the AKT/mTOR signaling pathway by blocking the activation of AKT and the downstream targets FKHR and GSK-3β. The effect of (6)-shogaol on the growth of cultured human and mouse prostate cancer cells was investigated [101,102] and the compound was found to reduce survival and induce apoptosis of both cell species. Mechanistic studies revealed that (6)-shogaol inhibited STAT3 and NFκB activation. The compound also slowed tumor growth in an allograft model. Liu et al. [102] demonstrated that 15 μM (6)-shogaol treatment induced apoptosis and G2/M phase arrest in HeLa cells and caused the differential expression of 287 proteins. 14-3-3 signaling was found to be a predominant canonical pathway.

(6)-Shogaol produced a strong inhibitory effect on NSCLC cell proliferation and anchorage-independent growth [103]. It induced cell cycle arrest and apoptosis, and inhibited AKT kinase activity by binding with an allosteric site of AKT. The induction of apoptosis corresponded with the cleavage of caspase-3 and caspase-7. In addition, it reduced the constitutive phosphorylation of STAT3 and decreased the expression of cyclin D1/3. Moreover, intraperitoneal administration of [6]-shogaol inhibited the growth of NSCLC cells as tumor xenografts in nude mice and suppressed the expression of Ki-67, cyclin D1 and phosphorylated AKT and STAT3 in the tumors.

(6)-Shogaol treatment of human umbilical vein endothelial cells or rat aortas attenuated the formation of tubes [96].

### 2.6. Cellular and Molecular Activities of Paradols

There have been some reports evaluating the chemoprevention qualities of (6)-paradol. Surh et al. [39] found anti-tumor-promoting properties of (6)-paradol. Topical application prior to TPA-attenuated skin papillomagenesis initiated by DMBA in female ICR mice. The substance also inhibited the tumor-promoter-stimulated inflammation, TNF-α production, and activation of epidermal ODC. Another study showed that (6)-paradol caused a significant decrease in the incidence and multiplicity of mouse skin tumors caused by the DMBA/TPA initiation/promotion protocol [104]. (6)-paradol also suppressed superoxide production stimulated by TPA in differentiated HL-60 cells.

Suresh et al. [105] and Mariadoss et al. [106] evaluated the chemopreventive potential of (6)-paradol on DMBA-induced tumors in the hamster buccal pouch. OSC carcinomas were induced by painting with DMBA three times a week for 14 weeks. Oral administration of 30 mg/kg of (6)-paradol on alternate days from DMBA painting for 14 weeks significantly reduced tumor formation. Altered expression of apoptotic-associated genes; p53, bcl-2, caspase-3, and TNF-α was observed compared to the DMBA-only animals.

Bode et al. [107] reported that AP-1 activation was blocked by (6)-paradol, thus inhibiting EGF-induced mouse skin cell transformation. (6)-Paradol induced apoptosis in oral squamous carcinoma cells in a dose-dependent manner through a caspase-3-dependent mechanism [108].

### 2.7. Cellular and Molecular Activities of Zingerone

Recently, there has been interest in zingerone as a chemopreventive compound. Bae et al. [109], investigated zingerone using a mouse skin tumor model and found that it suppressed tumor progression and tumor angiogenesis. Zingerone also inhibited the angiogenic activities of endothelial cells. The activities of MMP-2 and MMP-9 were decreased in the tumor cells. Investigators [110] also showed that it has anti-mitotic effects against human neuroblastoma cells and inhibits cellular viability of those cells in prometaphase. Zingerone-treated neuroblastoma cells also showed decreased cyclin D1 expression and induced cleavage of caspase-3 and PARP. Furthermore, zingerone inhibited the growth of neuroblastoma tumors in BALB/c mice.

Ganaie et al. [111] demonstrated the protective effect of zingerone against experimental colon carcinogenesis. Their results reveal that DMH-treated rats that received zingerone exhibited elevated ROS, increased activity of cytochrome P450 2E1 and the serum marker CEA, with a lowered number of ACF. There was also decreased expression of inflammatory proteins. Nrf-2 was downregulated in the tumors of the zingerone-treated animals with an accompanying reduction of the activities of cytochrome P450 2E1 and CEA. In addition, NFkB, COX-2, iNOS, PCNA, and Ki-67 were suppressed, as were the levels of IL-6 and TNF-α.

### 2.8. Cellular and Molecular Activities of α-Zingiberene

α-Zingiberene is primarily found in the peel of the rhizome [112]. It has strong anti-proliferative activity for human colon carcinoma cells in vitro. α-Zingiberene treatment produced nucleosomal DNA fragmentation in uterine carcinoma cells and the percentage of sub-diploid cells increased in a concentration-dependent manner. Further, the cells exhibited hallmark features of apoptosis as mitochondrial cytochrome c was released and CAS-3 was activated [113].

## 3. Bitter/Shampoo Ginger

The fresh rhizome has been used as a folk medicine for centuries [114]. However, the use of the whole rhizome parts/powders or crude extracts of the plant for cancer chemoprevention have not been reported. Zerombone (Figure 2) is the major bioreactive compound in the rhizome of the plant. Its cellular metabolism and biodistribution/pharmacokinetics have not been reported. The cancer chemopreventive studies and the molecular actions of zerumbone are reviewed below.

### 3.1. Initiation/Promotion Studies

The anti-initiating and anti-promoting activities of zerumbone in mouse skin carcinogenesis were evaluated using a conventional two-stage carcinogenesis model [115]. A single topical pretreatment of zerumbone 24 h before application of DMBA suppressed tumor incidence by 60% and the number of tumors per mouse by 80%. Zerumbone enhanced the mRNA expression level of MnSOD, GPx1, GSTP1 and NAD(P)H quinone oxidoreductase. Further, it diminished TPA-induced COX-2 protein expression and phosphorylation of ERK1/2. Histologic examination revealed that pretreatment with zerumbone suppressed inflammation by suppressing leukocyte infiltration into the lesion. They also reported that topical application of zerumbone onto skin induced the activation of Nrf2 and expression of HO-1. Moreover, zerumbone suppressed TPA-promoted cell transformation and the intracellular accumulation of ROS.

The effects of feeding zerumbone on the azoxymethane-dextran sulfate initiation/promotion carcinogenesis protocol of ACF development in mice were investigated [116]. Zerumbone caused reduction in the frequency of ACF at a dose of 0.05%. In addition, the compound significantly reduced expression of COX-2 and prostaglandins in colonic mucosa along with suppression of colonic inflammation, inhibition of proliferation, induction of apoptosis, and suppression of NFκB and HO-1 expression in the tumors that developed.

The chemopreventive effects of zerumbone were assessed in rats induced to develop liver cancer using the DEN/AAF initiation/promotion model [117]. The animals received intraperitoneal zerumbone injections beginning on week 4 post-DEN injection. Serum ALT, AST, AP and αFP were lower in the zerumbone-treated rats, indicating a protective effect on the liver. There was also reduction in hepatic tissue GSH concentrations. PCNA measured in the liver by immunohistochemistry in the zerumbone-treated animals was lowered, and there were high numbers of apoptotic cells. Zerumbone treatment also increased Bax and decreased Bcl-2 protein expression in the livers.

The chemoprevention properties of zerumbone were investigated using female Balb/c mice exposed prenatally to diethylstilbestrol [118]. Zerumbone resulted in the regression of cervical intraepithelial neoplasia, abundant apoptotic cells, and modulation of Bax and Bcl-2 protein expressions.

Using the A/J mouse lung carcinogenesis protocol [119], it was found that zerumbone feeding inhibited the multiplicity of lung adenomas in a dose-dependent manner, along with inhibition of proliferation, induction of apoptosis, and suppression of NFκB and HO-1 expression in the tumors that developed.

### 3.2. Blood-Borne Cells

Kim and Yun [120] found that zerumbone (1–10 μM) inhibits the secretion of pro-inflammatory cytokines and the induction of NFκB by LPS-activated macrophages. In addition, zerumbone significantly inhibited mRNA and protein levels of TLR-2/4, and the expression of MyD88 protein. Moreover, zerumbone down-regulated the expression of HDAC genes. Haque et al. [121] examined the effects of zerumbone on inflammatory mediated MyD88-dependent NFκB/MAPK/PI3K-AKT signaling pathways in LPS-stimulated macrophages. The compound suppressed up-regulation of the pro-inflammatory mediators TNF-α, IL-1β, PGE_2_, and COX-2, and downregulated the phosphorylation of NFκB (p65), IκBα, and IKKβ. Zerumbone also caused attenuation of AKT, JNK, and ERK expressions, and p38 MAPKs phosphorylation in a concentration-dependent manner. Zerumbone diminished the expression of TLR4 and MyD88 as well.

Murakami et al. [122] found that zerumbone suppressed TPA-induced superoxide anion generation from NADPH oxidase enzymes in human promyelocytic leukemia cells. The compound exhibited reduced expressions of iNOS and COX-2, together with diminished release of TNF-α by combined LPS- and INF-γ-stimulation of macrophages. Furthermore, zerumbone inhibited growth of leukemia cells in a time- and concentration-dependent manner. Their cell cycle was arrested at the G2/M phase and the cyclin B1/cdk1 protein levels were decreased. Zerumbone caused cell growth inhibition with an IC50 value of 3.5 μg/mL for chronic myeloid leukemia cells [123]. Cytotoxicity was exemplified by DNA damage and pro-caspase-3, -9 activation-associated apoptosis. Zerumbone did not affect the growth of normal human peripheral blood lymphocytes.

Zerumbone was reported [124] to be cytotoxic to Jurkat cells in a dose- and time-dependent manner. The compound did not affect the growth of normal human peripheral blood mononuclear cells. Zerumbone arrested Jurkat cells at the G2/M phase of the cell cycle. The antiproliferative effect was through the intrinsic apoptotic pathway via the activation of caspase-3 and -9. In another study [125], the antitumor effects of zerumbone were measured in tumorigenic lymphoid cells, and lymphoid tumor cell-bearing CDF mice. The results showed that the zerumbone induced DNA fragmentation in the tumor cells in vitro, and prolonged the life of the tumor-bearing mice.

Xian et al. [126] demonstrated that zerumbone suppressed the proliferation of promyelocytic leukemia cells by inducing G2/M cell cycle arrest followed by apoptosis. Treatment of the cells was associated with a decline of cyclin B1 protein. Furthermore, zerumbone-induced apoptosis was initiated by the expression of Fas/Fas Ligand, concomitant with the activation of caspase-8. Zerumbone was also found to induce the cleavage of Bid, a mediator that is known to connect the Fas cell death receptor to the mitochondrial apoptosis pathway. The compound also induced the cleavage of Bax and Mcl-1 proteins.

### 3.3. Digestive Cells

Murakami et al. [127] examined the effect of zerumbone on the expression of proinflammatory genes and found it reduced IL-1α, IL-1β, IL-6, and TNF-α expression in dose- and time-dependent manners. Elucidation of whether zerumbone potentiates TRAIL-induced apoptosis in human colon cancer cells was investigated by Yodkeeree et al. [128]. They found that it caused up-regulation of the TRAIL receptors DR4 and DR5. Induction of both DRs was abolished by GSH, suggesting a critical role of high levels of ROS induction by zerumbone. Zerumbone also induced p53. Singh et al. [129] showed the molecular docking of zerumbone to the TNF-α protein. The docked complex was validated by showing that the α-β unsaturated carbonyl scaffold of zerumbone is the important moiety for the blocking activity.

Zerumbone exhibits antibacterial action against *Helicobacter pylori* in vitro [130]. This activity suggests that it might be useful for chemoprevention of gastric cancer. Zerumbone effects against angiogenesis were elucidated using gastric cancer cells [131]. The expression of VEGF by the cells, both in the basal state and following zerumbone treatment, was investigated, as well as cell proliferation and NFκB activity. VEGF expression and NFκB activity were inhibited by zerumbone. Further, the compound inhibited angiogenesis measured by quantifying tube formation by HUVEC cells co-cultured with fibroblastic and gastric cancer cells. Shamoto et al. [132] evaluated the effects of zerumbone on pancreatic-tumor-cell-induced angiogenesis. The compound inhibited mRNA expression and protein secretion of angiogenic factors and NFκB activity. HUVEC cell tube formation was enhanced by coculture with the pancreatic cells, and this activity was inhibited by zerumbone. Research was carried out to investigate whether zerumbone produces anticancer effects on pancreatic carcinoma cells [133]. The data showed a concentration- and time-dependent inhibitory effect on cell viability and induction of apoptosis. ROS production was increased by about 149% in cells treated by zerumbone and the expression of p53 protein was up-regulated.

Exposure of cultured rat normal liver epithelial cells to zerumbone resulted in induction of GST and nuclear localization of Nrf2. Zerumbone showed antiproliferative activity for human liver cancer cells [134]. DNA fragmentation occurred and the level of apoptosis increased in a time-course manner. The apoptotic process triggered by zerumbone involved up-regulation of the pro-apoptotic Bax protein and suppression of anti-apoptotic Bcl-2 protein. These latter changes were independent of p53, since zerumbone did not affect the levels of p53, even though this protein exists in a functional form in these cells. Nakamura et al. [135] investigated phase II detoxification enzymes induction in cultured rat normal liver epithelial cells by zerumbone. Exposure resulted in an increase in GSH. Zerumbone induced nuclear localization of Nrf2 that binds to the antioxidant response element (ARE) of phase II enzyme genes, suggesting that it is a potential activator of the Nrf2/ARE-dependent detoxification pathway, as exemplified by enhanced expression of several Nrf2/ARE-dependent phase II enzyme genes, including GCS, GPx1, and HO-1.

The inhibitory potential of zerumbone against hepatocellular carcinoma was studied by Wani et al. [136]. Zerumbone inhibited proliferation and survival of hepatic cancer cells in a dose-dependent manner by arresting them at the G_2_/M phase, inducing apoptosis, and inhibiting the PI3K/AKT/mTOR and STAT3 signaling pathways. Tracing glucose metabolic pathways revealed a reduction in glucose consumption and reduced lactate production, suggesting glycolytic inhibition by zerumbone. In addition, the compound impeded the shunting of glucose-6-phosphate through the pentose phosphate pathway. Zerumbone treatment also suppressed subcutaneous and orthotopic growth and lung metastasis of hepatocarcinoma cells injected into immunocompromised mice. Samad et al. [137] showed that zerumbone inhibited proliferation and suppressed cell migration of hepatocarcinoma cells in a dose-dependent manner. Lv et al. [138] also reported that hepatocarcinoma cells treated with zerumbone exhibited induced apoptosis and their invasion and metastasis capabilities were suppressed in a dose-dependent manner. Zerumbone treatment resulted in a dose-dependent induction of apoptosis and cell cycle arrest at the G2/M phase, along with an increased expression of p27, cytochrome c, caspase-3 and-9, and Bcl-2-associated X, and decreased expression of cdk 1, cyclin B1, BCL, FAK, RhoA, ROCK, and MMP-2 and -9. In addition, the phosphorylation of p38 MAPK and ERK1/2 was up-regulated in a dose-dependent manner.

Wang et al. [139] determined the actions of zerumbone on cultured human gastric cancer cells. They observed that it inhibited their growth in a dose-dependent manner with the induction of apoptosis. Treatment with zerumbone downregulated Cyp A and Bcl-2 levels, upregulated Bax levels, and caused cytochrome c release.

### 3.4. Breast Cells

Han et al. [140] investigated the effect of zerumbone on IL-1β-induced breast cancer cell migration and invasion by analyzing the levels of IL-8 and MMP-3 mRNAs and found that the compound decreased both migration and invasion. The inhibitory effect of zerumbone on IL-1β-induced signatures correlating with cell invasion and signaling activation were also investigated by Jeon et al. [141]. Their results showed that zerumbone decreased the basal IL-1β expression level, and the phosphorylation level of NFκB, but did not suppress IL-1β-induced AKT phosphorylation. Zerumbone also upregulated TNF-related apoptosis and induced expression of the TRAIL ligand. Fatima et al. [142] observed that the compound is able to penetrate into the hydrophobic pockets of TNF-α, IKKβ and NFκB proteins with significant binding. Thus, zerumbone can exert its apoptotic activities by directly affecting cytoplasmic proteins.

Zerumbone was shown by Sung et al. [143] to down-regulate the CXCR4 receptor, which is expressed in breast tumors and mediates the homing of tumor cells to specific organs that express its ligand, CXCL12. Knockdown of Notch2 protein causes inhibition of cell migration, thus, the effects of zerumbone on Notch2 expression were investigated [144] and exposure of breast cancer cells to the compound resulted in increased transcriptional activation of Notch2. Kim et al. [145] investigated the role of zerumbone on the regulatory mechanisms of CD44 expression and found that its expression was increased by EGFR ligands. Zerumbone down-regulated the induction of CD44 expression by exposure to EGF. Finally, it was shown that EGF-induced phosphorylation of STAT3 was suppressed by zerumbone treatment. The resultant cell cycle arrest (G2/M) and apoptosis induction was associated with down-regulation of cyclin B1, Cdk 1, Cdc25C, and Cdc25B [146]. Zerumbone-treated breast cancer cells exhibited a robust activation of both Bax and Bak. In vivo growth of orthotopic xenograft breast tumor cells was significantly retarded by zerumbone administration with apoptosis of the tumor cells [147].

### 3.5. Skin Cells

Yang et al. [148] evaluated the protective effects of zerumbone against UVA using HaCaT cells and mouse epidermis. Zerumbone pretreatment suppressed UVA-induced cell death and LDH release in a dose-dependent manner. UVA-induced ROS production, DNA single-strand breaks, and apoptotic DNA fragmentation were reversed by zerumbone. Zerumbone-mediated cytoprotective properties were associated with increased nuclear translocation of Nrf2 accompanied by induction of the HO-1 and γ-GCLC genes. The effect of zerumbone on malignant melanoma cells was evaluated by Wang et al. and Yan et al. [149,150]. They determined that zerumbone was cytotoxic, induced apoptosis, down-regulated Bcl-2 gene mRNA and protein levels, up-regulated Bax and cytochrome c gene and protein levels, and activated Caspase-3.

### 3.6. Other Cells

The level of cellular internalization of labeled zerumbone was investigated [151]. HeLa cells exhibited selectively greater cellular uptake, as compared to normal cells. Zerumbone caused a mitotic block with the cells exhibiting bipolar spindles. Docking analysis indicated that tubulin is a principal target of zerumbone. The compound also inhibited microtubule assembly and induced an increase in MPM-2 expression in prostate cells [152]. It also caused an increase in the phosphorylation of Bcl-2 and Bcl-xL. Furthermore, zerumbone increased Mcl-1 protein expression, down-regulated Cdc25C, induced mitochondrial damage and endoplasmic reticulum stress, and upregulated the expression of GRP-78 and GADD153.

Park et al. [153] also investigated the role of zerumbone on angiogenesis using HUVEC cells. It inhibited cell proliferation, migration, and tubule formation, and phosphorylation of VEGFR-2 and FGFR-1, key regulators of endothelial cell function and angiogenesis. Osteopontin (OPN) induces cell invasion through inactivation of cofilin, which is mediated by the FAK/AKT/ROCK pathway in NSCLC cells. Zerumbone suppressed OPN-induced cell invasion and protein expression of ROCK1, as well as the phosphorylation of LIMK1/2 and cofilin. It also decreased OPN-induced lamellipodia formation on the cells [154].

Abdelwahab et al. [155] analyzed the effects of zerumbone on ovarian and cervical cancer cell lines and observed growth inhibition, arrest at G2/M phase, and apoptosis. Zerumbone significantly decreased the levels of IL-6 secretion by both cancer cells. Jorvig and Chakraborty [156] reported that zerumbone caused cell cycle arrest of prostate cancer cells at the G0/G1 phase followed by apoptosis, and reduced expression of multiple cancer-associated genes, including cyclin D1, IL-6, COX2, ETV1, and JAK2.

RCC cells exposed to zerumbone underwent apoptosis, as indicated by cell viability inhibition and DNA fragmentation [157]. Zerumbone activated caspase-3 and caspase-9 and downregulated Gli-1 and Bcl-2. Persistent activation of STAT3 is a characteristic feature of RCC cells and Shanmugam et al. [158] observed that zerumbone suppressed activation of STAT3 in a dose- and time-dependent manner. The suppression was mediated through inhibiting the activation of the upstream kinases c-Src, JAK1, and JAK2. In contrast, zerumbone induced the expression of SHP-1. Finally, when administered i.p., zerumbone inhibited the growth of human RCC xenograft tumors.

The CXCR4-RhoA and PI3K-mTOR signaling pathways play crucial roles in tumorigenesis of OSC cells. Zainal et al. [159] examined a panel of OSC cells and found that zerumbone inhibited their proliferation and induced cell cycle arrest and apoptosis. On the other hand, growth of normal keratinocytes was unaffected. In addition, zerumbone treatment inhibited migration and invasion of OSC cells, with concurrent suppression of CXCR4 receptor protein. RhoA activity was also reduced. The effects of zerumbone on the proliferation and apoptosis of ESCC cells and on their p53 and Bcl-2 expression levels were reported by Ma et al. [160]. The rate of proliferation inhibition was increased in a concentration-dependent manner and apoptosis occurred. The mRNA expression level of p53 in the treated cells was increased and that of Bcl-2 was decreased. Wang et al. [161] reported that zerumbone inhibited the migration of human ESC cells and caused Rac1 protein down-regulation in a dose- and time-dependent manner.

Treatment of cultured NSCLC cells with zerumbone caused a dose-dependent inhibition of cell viability and induced mitochondrial-associated apoptosis, as evidenced by loss membrane potential, release of cytochrome c, and activation of caspase-9 and -3. Also, there was increased p53 and Bax expression and ROS concentration in the zerumbone-treated cells, and down-regulation of p53 or the scavenging of ROS antagonized the pro-apoptotic action of the compound [162]. Zerumbone also induces apoptosis in human glioblastoma cells, along with caspase-3 activation and decreased IKKα phosphorylation levels in a time-dependent manner. Zerumbone also diminished AKT phosphorylation levels [163].

## 4. Conclusions and Future Research

The above demonstrates that *Zingiber officinale* Roscoe (true ginger) and its bioreactive constituents negatively affect carcinogenesis of multiple cell types via multiple molecular mechanisms. The same is true for zerumbone, which is extracted from *Zingiber zerumbet* Smith (bitter/shampoo ginger). In fact, both agents antagonize tumorigenesis in initiation/promotion carcinogenesis models, and tumor cell growth as xenotransplants.

Most of the hallmarks of chemoprevention, including depression of inflammation (NFκB and COX-2), induction of phase II detoxication enzymes (Nrf2), induced genomic instability (tubulin interactions), altered gene pathway expressions (increased expression of p27, cytochrome c, caspase-3 and-9, and Bcl-2-associated X, and decreased expression of cdk1, cyclin B1, BCL, FAK, RhoA, and ROCK), changes in microRNA translation, decreased proliferation and cell apoptosis (p53), impeded tumor cell invasion (Notch2), and neo-vascularization are affected by these compounds. Interestingly, the active constituents of both ginger species have the property to bind to and inhibit the activity of cytoplasmic proteins.

Ginger constituents are selective for tumor cells, as normal and non-tumorigenic cells are not affected. The mechanism(s) of this selectivity for gingerols, shogaols, and paradols appears to be differential activation of the glycosylated metabolites by higher levels of β-glucuronidase in the tumor cells, whereas zerumbone may discriminate because of cell-type-specific differential up-take. However, more research should be undertaken to further elucidate these mechanisms, as the information gleaned would further help in the understanding of carcinogenesis and chemoprevention processes. Another interesting avenue of research would be elucidating why some of the tumor cells are resistant to these compounds, as they do not completely eradicate all of the cancer lesions.

Biodistribution studies of zerumbone are necessary. Further, relatively few chemoprevention animal dose–response studies have been reported. Thus, more investigations into the timing of application of the active compounds in initiation/promotion protocols should be carried out. Animal studies with true ginger compounds combined with zerumbone would be interesting. However, it might be predicted that these studies would not show additive/synergistic effects, as they utilize comparable mechanisms. On the other hand, the compounds may have different pharmacokinetics and they may augment each other when used in combination.

A few human studies have been reported using true ginger extract, but none have been undertaken with zerumbone. As numerous anti-cancer markers of activity are known, after more animal studies are completed [164], more human trials should be undertaken, especially using the purified active compounds.

## Figures and Tables

**Figure 1 molecules-24-02859-f001:**
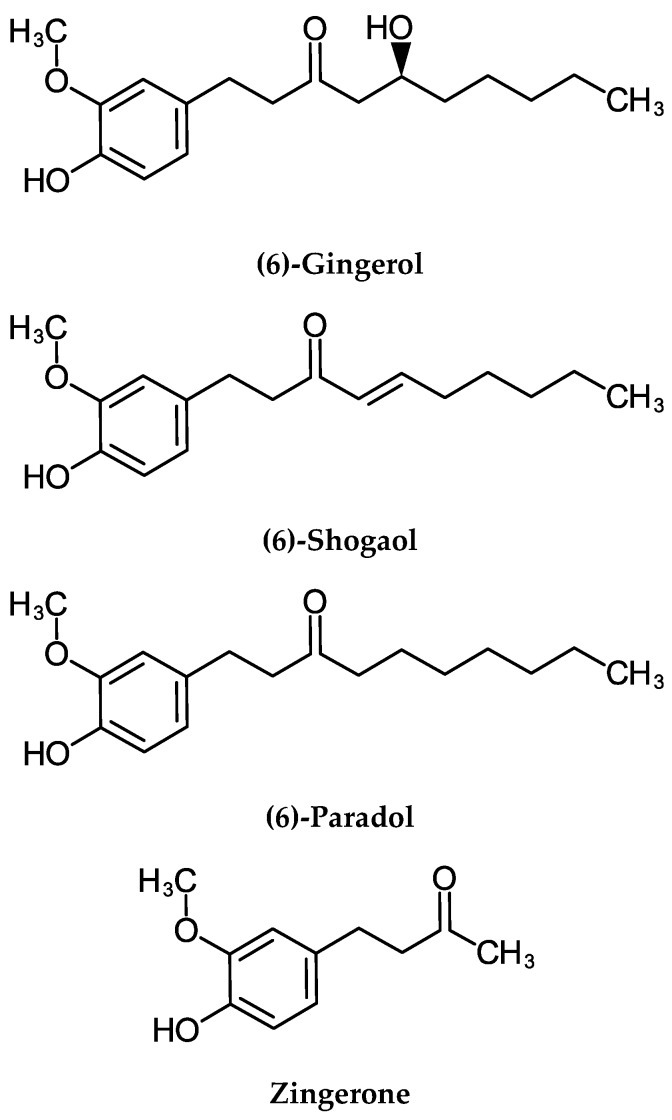
Chemical structures of ginger rhizome (*Zingiber officinale*) constituents.

**Figure 2 molecules-24-02859-f002:**
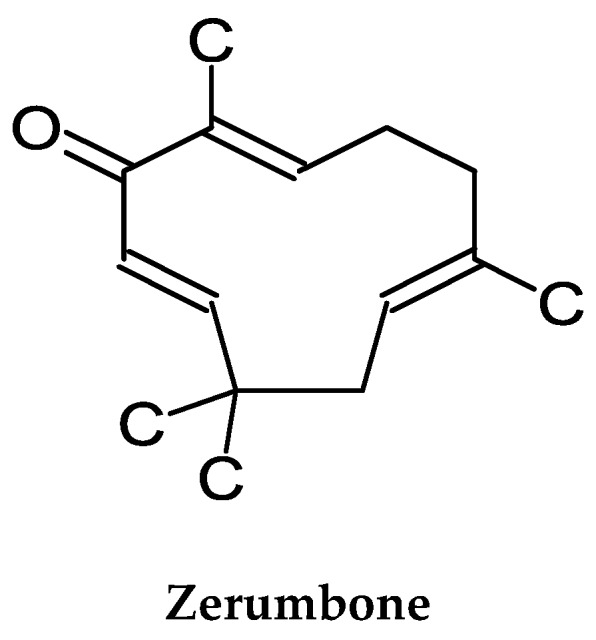
Chemical Structure of Zerumbone.

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
