# Peer review of "Gingers and Their Purified Components as Cancer Chemopreventative Agents"

_molecules, 2019, doi:10.3390/molecules24162859_

Round 1
Reviewer 1 Report
Manuscript molecules-564302 reviewed the chemo-preventative activities of ginger extracts (true ginger and bitter/shampoo ginger) and their components/compounds (Gingerol, Shogaols, Zerumbone, etc.). The topic is interesting, and the summary is helpful to discover new molecules/drugs for the potential chemoprevention. Thus, this manuscript is recommended to publish on Molecules after minor revision.

Author Response
Re: Manuscript ID - molecules-564302
My co-author Dr. Stoner (Guest Editor of the Special Issue: Natural Products for Cancer Chemoprevention) and I appreciate the work of the reviewers to improve our publication. However, we feel that the outline we have used to review the enormous literature better conveys the data than does the outline suggested by reviewer 1. Our outline allows the reader interested in, for example, breast cancer to quickly read the specific data relevant their disease interest.
Both reviewers 2 and 3 suggest that we add tables. We thought about this when writing the MS and rejected it because there was an enormous amount of data to present and to discuss. Adding tables would have markedly increased the length of a paper which, in all probability, is already too lengthy because it would require that we discuss all data presented in the table as we have done in our version. Further the tables would be more complex than is the written text.
Quite honestly, at the beginning, we did not realize the extent of the database on ginger and its components and we hope that the reviewers will find our paper acceptable with only the minor suggested modifications.
Thank you for your consideration.
John F. Lechner and Gary D. Stoner
Reviewer 2 Report
Major comments:
1. In my opinion, the main key sentence is in the final one in abstract. Please provide more information related to the review. If need, some sentences above can be considered to remove.
2. Except for the chemical structure, the authors provide the text only description. I suggest the authors to generate a brief Table to summary these descriptions. Or, at least some sections may need to provide a Table or figure. It will let the readers have a brief view immediately.
Minor comments:
1. line 338: Nazim and Park show special colored letters. Please change to black.
Author Response

(The authors gave the same response as above.)

Reviewer 3 Report
In this review, John F. Lechner et al discuss whereby the chemoprevention activities of gingers antagonize cancer development. In general, this review is detailed and logical. However, the whole article is made up of text, and it is suggested that the authors can summarize the issues they want to explain by tables or diagrams.
Author Response

(The authors gave the same response as above.)

Round 2
Reviewer 2 Report
I accept the authors' response.